# Personal and workplace factors influencing the resilience of nurses caring for women with cervical cancer in a resource-constrained setting in Ghana

**Jennifer Oware, Merri Iddrisu, Kennedy Dodam Konlan** *, **Gladys Dzansi**

Department of Adult Health, School of Nursing and Midwifery, University of Ghana, Legon, Greater Accra Region, Ghana

* kdkonlan@ug.edu.gh, kennedy.konlan@gmail.com

**Data Availability Statement:** Data cannot be shared publicly due to certain employee information and confidential issues which will be compromised if the data is published online. Data

## Abstract

### Introduction

Cervical cancer has been identified as the fourth most common form of malignancy affecting and killing women globally. Nurses caring for cervical cancer patients are exposed to emotional and psychological distress due to late presentation and the burden of care. Resilience has been identified as one of the effective ways of helping nurses to cope well with the stress of oncology nursing, but this remains undetermined in Ghana

### Aim

This study explored personal and workplace factors influencing the resilience of nurses caring for women diagnosed with advanced cervical cancer (stage III and IV) in a resource-constrained setting in Ghana

### Method

Using a qualitative approach, we recruited twenty nurses and midwives who had worked for a year and above caring for advanced-stage cervical cancer patients at the national referral hospital in Ghana. We conducted in-depth interviews between July, 2022 to September, 2022 which were audio-taped with participants' consent. Transcription was done verbatim, and analysis conducted using thematic analysis approach with the aid of NVivo 10.0.

### Results

The results revealed experience as a safety toolkit, inherent desire to help/care for the patient, emotional numbness and maintaining professional outlook as personal factors influencing resilience among the participants. Regarding the workplace factors influencing resilience, we identified the main theme of demands of caregiving for advanced cervical cancer patients with the following sub-themes; severity of cases managed, nature of care rendered, activities of care given, reshuffling, schedules and gender mirroring as an exacerbator of psychological suffering.

are available from the Institutional Review Board of the Korle Bu Teaching Hospital (Contact via; Hotlines: +233-(302) 739510 (Main) / (244)-406700 / Whatsapp: (243) 407809 | Email us: info@kbth.gov.gh | GA-221-1570)

**Funding:** The author(s) received no specific funding for this work.

**Competing interests:** The authors have declared that no competing interests exist.

**Abbreviations:** CNM, Cervical Cancer Nurse or Midwife; KBTH, Korle Bu Teaching Hospital; KATH, Komfo Anokye Teaching Hospital; SSA, Sub-Saharan Africa.

## Conclusion

Resilience among nurses and midwives caring for terminally ill cervical cancer patients is influenced by longer years of service, intrinsic motivation to work as a nurse, and the defense strategy of emotional numbness and professionalism at the individual level. Also, the huge demand of caregiving serves as a major workplace factor affecting the resilience of nurses and midwives. We recommend strategies such as regular ward conferences and in-service trainings aimed at enhancing job-experience, inherent desire to render care and professionalism be adopted in resource-constrained settings to improve nurses' resilience. In addition, political actors and management of hospitals must prioritize allocation of resources for advanced cervical cancer care with particular focus on providing more specialized nurses and midwives.

## Introduction

Cervical cancer refers to malignancy affecting the cervix/neck of the uterus [1] and has been identified as the fourth most common form of malignancy affecting and killing women globally [1–3]. According to the Global Cancer Observatory [2, 3], the worldwide incidence of cervical cancer stands at 3.1% of all cancer cases in 2020 and this translates to over six hundred thousand (604,127) new cases of cervical cancer each year [3]. The literature (2, 3) states that the African continent accounted for almost one hundred and twenty thousand (117,316) of the global cervical cancer cases in 2020 and this constituted almost twenty percent (19.4%) of the global incidence. In West Africa, almost twenty-eight thousand (27,806) cases of cervical cancer were diagnosed in hospitals in 2020 [3] and this constituted about 4.6% of the global incidence [4] with most cases unreported and undiagnosed [3, 4]. In Ghana, almost two thousand and eight hundred (2,797) new cases of cervical cancer were diagnosed [4, 5] and this constituted 11.6% of all cancer cases diagnosed in 2020. Therefore, cervical cancer is ranked as the third (3rd) cancer in Ghana after breast and liver cancers [4, 5]. In terms of mortality, cervical cancer accounted for over three percent (3.3%) of all cancer-related deaths worldwide with over three hundred thousand (341, 831) deaths in the year 2020 [3, 5]. Africa accounted for almost eighty thousand (76, 745) of the global deaths and this represented over twenty percent (22.45%) of the global deaths [1–3]. In Ghana, almost two thousand (1,699) women die from cervical cancer each year with most of the deaths unreported [3, 5].

Resilience among nurses and midwives in cancer care has been identified as one of the cardinal tools in improving patient outcomes [6–8] due to its effect on burnout reduction [9]. Caring for cancer patients, therefore, requires highly resilient nurses who will be able to cope well with the stressors of oncology nursing [8–10]. Several studies have reported that nurses working in oncology environments face a lot of psychosocial crises such as burnout, emotional exhaustion, depersonalization, and compassion fatigue [4, 5, 6]. These have been found to be related to extensive workload resulting from the burdensome nature of patients' illness, prolonged exposure to patient's suffering, pain and frequent death of cancer patients which affects their psychosocial wellbeing [5–9]. Furthermore, studies have revealed that nurses caring for cancer patients often ignore or pay less attention to the emotional crises they encounter thinking it is normal which with time affects their work output and resilience [8–10].

Nurses' psychosocial wellness and resilience are impacted by their everyday interactions with cervical cancer patients who are terminally ill and dying [4, 10], as well as by their

frequent exposure to patients' pain and death [7, 8]. Failure to identify the connection between work and health as well as long-term exposure to work-related stressors among nurses has been identified as the highest contributing factor to burnout, emotional exhaustion, depersonalization, and compassion fatigue among nurses [6, 10]. There is a consensus in the literature from Europe that providing care for patients with cervical cancer has an emotional influence on midwifery and nursing staff working in an oncology workplace [3, 7, 10]. The pressures also have an impact on their degree of job satisfaction and nurses' view of how helpful their workplace is to them [3, 7]. However, there is a dearth of literature related to the personal and job-related factors influencing the resilience of nurses caring for women diagnosed with advanced cervical cancer particularly in resource-constrained settings in sub-Saharan Africa.

## Aim

This study explored personal and workplace factors influencing the resilience of nurses caring for women diagnosed with advanced cervical cancer (stage III and IV) in a resource-constrained setting in Ghana.

## Methods

### Study design

We adopted an exploratory descriptive approach to qualitative research as the study design to elicit in-depth information related to the aim of the study [11]. A qualitative study involves the collection, analysis and interpretation of non-numerical data in the form of descriptions of personal experiences narrated by the study participants [11]. The qualitative approach was chosen for this study because it seeks to understand how nurses and midwives subjectively appraise their daily experiences in the care of patients with advanced cervical cancer and how those experiences influence their psychosocial well-being and resilience. Exploratory research is normally used in studying phenomena that have not received much previous investigation and are meant to be a preliminary study to establish the basis for further studies.

### Setting

The study was conducted at the Korle Bu Teaching Hospital (KBTH) in Accra, Ghana. With nearly 2,000 beds and 17 clinical, diagnostic, and specialist departments/units—including cancer treatment—the KBTH is Ghana's largest health facility and main teaching hospital [12, 13]. The hospital, which opened on October 9, 1923, is now the third largest in Africa. Every day, almost two thousand (2000) patients visit the hospital with about two hundred and fifty (250) of these patients admitted daily. The hospital serves as referral hospital for the sub-region and provides a wide range of medical services, including general medicine, paediatrics, obstetrics, gynaecology, pathology, laboratory, radiology, anaesthesia, surgery, polyclinic, accident centre, surgical/medical emergency, and pharmacy. Moreover, it has a financial department, engineering laboratory, administrative offices, and a pharmacy [13].

   Specifically, the study took place in the Obstetrics and Gynaecology Department of the Hospital. Patients diagnosed with advanced cervical cancer are admitted to the gynaecology unit for management. Data for the study was therefore collected among nurses and midwives at the gynaecology unit of the KBTH where patients diagnosed with advanced cervical cancer were receiving treatment. Most of these patients with advanced cervical cancers were referred cases from all over the Country and the West African sub-region. Information from the Unit Heads of the various wards of the Gynaecology Unit of the KBTH revealed that advanced cervical cancer cases constituted almost seventy percent of all cervical cancer cases being managed

on the Unit due to the poor reporting culture of most patients. This made the setting appropriate for the study as it focused on nurses and midwives providing care for patients with advanced cervical cancer (stage III and IV).

## Study population and sample size determination

The target population for this study were nurses and midwives working at the gynecology unit of the KBTH where advanced cervical cancer patients in Ghana were being managed. The study used non-probability sampling technique of purposive sampling to select study participants for the data collection.

The participants were recruited if they met the following inclusion criteria: must be nurses and midwives caring for patients with advanced stage cervical cancer (stage III and IV); who had a current license as a registered nurse or midwife, with at least one year of professional nursing experience and had given care to a patient with cervical cancer in the previous twelve (12) months prior to the study.

We excluded nurses and midwives with less than twelve (12) months' work experience in the care of advanced cervical cancer patients. Also, nurses on national service, nursing students and auxiliary nurses (unprofessional nurses and midwives) were excluded from the study.

During the data collection, the researchers observed data redundancy after the sixteenth interview. We conducted further interviews and realized that after conducting four additional interviews after the sixteenth interview, no new or additional information was being elicited from the participants. Therefore, data saturation (the stage where no more additional information was elicited from the participants) was reached at the twentieth interview.

## Participants and selection criteria

The participants for this study were nurses and midwives providing nursing care for patients with advanced cervical cancer (stage III and IV) at the Korle Bu Teaching Hospital.

We selected the actual participants on weekdays from the gynecology Unit of the KBTH where patients diagnosed with advanced cervical cancer were being managed. We confirmed from the Unit Heads of the various wards of the Gynecology Unit of the KBTH who stated that advanced cervical cancer cases constituted almost seventy percent of all cervical cancer cases being managed on the Unit due to the poor reporting culture of most patients. Using the participant information sheet in English, we fully explained the purpose, risks, benefits of the study to the participants before obtaining their written informed consents.

## Data collection

We conducted in-depth interviews (IDIs) in English with the participants after obtaining voluntary written informed consent in English. The interviews were conducted by the 1st and 3rd authors with the supervision of the 2nd and last authors who have extensive experience with conducting interviews in qualitative studies. The data collection was done between July, 2022 and September, 2022. We initially contacted about sixty-four (64) nurses and midwives working in the Gynecology Unit who had met the inclusion criteria stated earlier but only forty-eight (48) who met the inclusion criteria expressed interests in participating after reading and understanding the participant information sheet which was in English (the official language for communicating in Ghana) and were willing to provide written informed consent.

The data collection was done at the nurses' restroom of the Gynecology Unit using a semi-structured interview guide (S2). This semi-structured interview guide allowed the participants to express themselves about the research area. This tool was developed by the researchers and pre-tested among a similar population (two patients with advanced cervical cancer) at the

Komfo Anokye Teaching Hospital in Kumasi, Ghana. The interview guide (S2) contained open-ended questions that allowed further probing to elicit relevant responses about the experiences of the respondents in the care of patients with advanced stages of cervical cancer. The first section of the instrument solicited data on the socio-demographic characteristics of the participants. Data collected included the age, highest educational qualification, years of experience, marital status and religion, number of children, and number of dependents. The second section focused on the individual characteristics of the nurses and how they affected their resilience and the third section focused on workplace characteristics and how they influenced the resilience of the participants.

The interviews were conducted in English language (the official language for communicating in formal settings in Ghana) at the rest room of the nurses and midwives in the gynecology unit where privacy was assured. An audio tape was used to record the interviews. The recorded audio files were transcribed verbatim by the 1[st] and 3[rd] author and thereafter thematic analysis was done. The interviews lasted 45–60 minutes for each participant. In addition, the 1[st] and 3[rd] authors who did the data collection under supervision of the 2[nd] and last authors between July, 20222 and September 2022, maintained a field notebook in which observations were recorded and this was used in the data analysis. During the interviews, all the participants were asked the same questions in the same format and the participants were asked to freely express themselves using further probing questions to elicit in-depth information.

## Quality control

In this study, we specifically put in place mechanisms to improve the data quality as expected in literature [11]. The interview guide was pre-tested at the Komfo Anokye Teaching Hospital in Kumasi, Ghana among similar participants. The pre-testing was conducted by the 1[st] and 3[rd] author with the supervision of the 2[nd] and last authors among two (2) nurses who met the inclusion criteria and were providing care to women with advanced cervical cancer at the Komfo Anokye Teaching Hospital in the Asanti Region of Ghana (a setting similar to the study setting). The pilot interviews which lasted between 20–35 minutes were audio taped. This exercise helped to ascertain the clarity of the questions in the interview guide and determined whether the questions sought to answer the research questions. Questions that were observed to be narrow and restrictive responses were modified to include the words "describe" or narrate to broaden the responses and to give the participants the opportunity to speak more on the topic [11]. During the data collection, the 1[st] and 3[rd] authors with the supervision of the 2[nd] and last authors adhered strictly to the methodological rigour described below and enhanced member checking with the participants to ensure that their views had been properly represented in the transcripts from the interviews [11]. The transcription of the audio-files which had been taped in English, the official language in the study setting was done by the 1[st] and 3[rd] authors with the supervision of the 2[nd] and last authors to ensure the quality of the data collected. Specifically, after the transcription of the audio-files, the 1[st] and 3[rd] author with the supervision of the 2[nd] and the last authors met the participants again and made them to read the transcripts as well as listen to their respective audio-files to be sure the transcripts reflected exactly the information on the recorded audio-files. This was done for all the twenty participants who took part in the study and they all stated that the transcripts reflected exactly the information in the audio files which had been recorded. At the end of the data collection, the audio transcripts in English were confirmed as verbatim transcripts of the audio files by all the participants.

## Data management

Interviews were audio recorded and the data was kept in a folder with a password. The interviews were transcribed verbatim from audio to text. The transcripts were reviewed with the audio recorded files several times to verify any discrepancies and we contacted the participants where there was a need for clarification. The participants were given codes based on their ward and their position among the participants. Participants' confidentiality was ensured by removing all identifying attributes from the data and replacing them with pseudonyms; such as cervical cancer nurse or midwife (CNM). All identifying attributes such as contact information were stored electronically in a password-protected laptop, likewise, consent forms, field notes and other important documents were kept under lock and key and the file was only accessible to the authors. The data was also stored on an external hard drive to prevent data loss

## Rigour

To ensure credibility, the interview guide was pretested among two (2) nurses who met the inclusion criteria and were providing care to women with advanced cervical cancer at the Komfo Anokye Teaching Hospital in the Asanti Region of Ghana (a setting similar to the study setting). This allowed the researchers to make necessary modifications to aspects of the interview guide that seemed not to elicit responses that were relevant to the objectives of the study. Additionally, a purposive sampling technique was employed to ensure only nurses and midwives rendering care to women diagnosed with advanced cervical cancer who could give a vivid account of their experiences were recruited. Probing and iterative questioning was also used to elicit responses from participants and situations where there were ambiguities in the responses; clarifications were sought from the participants. To achieve transferability, the research process was described in detail so that others can evaluate the applicability of data to other contexts and settings. Records of the transcribed interviews and the analysis, as well as the results of the study, were kept for audit trail and are in the custody of the 1st author. To ensure dependability in the study, a pre-tested semi-structured interview guide was used for all the interviews to ensure consistency in the line of questioning among the participants. Again, a detailed description of the study design, sampling method, data collection, and analyses were as well documented. To ensure confirmability, the context of data collection was documented in a field notebook during the interviews. This enhanced interpretation of the data during analysis to reflect the exact responses of the participants. The authors also bracketed their experiences as professional nurses to avoid any influence in the interpretation and analysis of the data.

## Data analysis

We performed the data analysis concurrently with the data collection with the aid of Nvivo 10.0. The 1st and 2nd authors adopted the thematic analysis approach to data analysis as recommended in literature [14–16] and performed thematic analysis under the guidance of the 2nd and last authors. The interviews that were audio recorded were transcribed word for word as soon as they were concluded. The verbatim transcripts and the field notes recorded in the diary were used in conducting the thematic analysis [14–16]. The analysis followed the procedure as outlined by Braun and Clarke [14–16]. The 1st and 2nd authors read the transcripts and field notes multiple times to become comfortable with the material and pick out notable sentences and phrases. Statements and phrases that were found to be similar were categorized into several folders and given unique names. To begin describing the individual factors, and workplace factors of the participants which influenced their resilience, the similar phrases were grouped into meaningful units to generate the initial codes that led to the development of

themes and sub-themes [16]. Direct quotations were captured in the analysis to support the themes by using specific quotes for each cervical cancer nurse or midwife (CNM). The themes generated from the data by the 1st and 3rd authors were confirmed by the 2nd and last authors as the themes emerging from the data.

### Ethics approval and consent to participate

This study was approved by the Institutional Review Board of the Korle Bu Teaching Hospital (Protocol Number: KBTH-IRB/00018/2022) (S1). The study was conducted in conformity with the Helsinki Declaration on Human Experimentation, 1964 with subsequent revisions, latest Seoul, October 2008. Permission was also sought from the head of gynecology unit before data collection. We provided a participant information sheet in English (the official language for communicating in Ghanaian formal settings) to all the participants who met the inclusion criteria to read prior to obtaining informed consent. The participant information sheet provided each eligible participant with information of his/her right to withdraw from the study at any time without suffering any negative consequences, information about the study objectives, benefits, risks among others. Written informed consent was also obtained from each participant before recruitment into the study. Names of the participants were not revealed in the study report and all information gathered from the study participants were treated confidentially as special codes were used to represent the responses of each participant.

## Results

The study had twenty (20) participants. All the participants had been caring for advanced cervical cancer patients for over a year. We analyzed the data based on the objective of the study which was to explore personal and work-related factors influencing resilience in nurses and midwives caring for women with advanced cervical cancer in Ghana.

The data analysis yielded some main themes and sub-themes. Table 1 presents the synthesis of the themes and sub-themes gleaned from the data (Table 1).

### Personal characteristics of nurses and midwives affecting resilience

Individual characteristics affecting resilience that were identified in this study included experience as a safety toolkit, inherent desire to help/care for the patient, emotional numbing, professional boundaries and self-motivation.

**Table 1. Synthesis of themes and sub-themes.**

| Major theme | Sub-theme |
| --- | --- |
| Personal factors | Experience as a safety toolkit |
|  | Inherent desire to help/care for the patients |
|  | Maintaining professional outlook |
|  | Emotional numbness |
| Demands of caregiving | Other cases being managed |
|  | Severity of cases managed |
|  | Nature of Care |
|  | Reshuffling |
|  | Gender mirroring as an exacerbator of psychological suffering. |

## Experience as a safety toolkit

Experience was identified as a safety toolkit for promoting resilience among the participants. They claimed their experiences working with advanced cervical cancer patients improved their ability to withstand the shocks and demands of the job which they claim was useful in promoting resilience. Some quotes of the participants are shown below:

*"You know after some years you gain experience, sometimes you notice one sign and you know what it means, what is about to happen or what is at stake. You notice something small or little change and you want to look out for this or that or you want to call your doctor early. You know that you have to be on guard. So, it has made work faster and then more effective and it has even made the care we give to the patient better because now you know what you are about"*. (CNM04)

*"This work is all about experience, the more you work on the job, the more you get to know what to do when you observe or see certain things happening to the patient"*. (CNM19)

*"We the older ones have been here for sometime now so we have developed more inward strength in relation to the work due to our personal experiences with the work and this helps us to cope better"*. (CNM 02)

Some participants described that their years of experience have made them strong to withstand stressful situations:

*"...Right from 2011 till now, I wasn't as tough as I am now. Yes, because there were little things that went on around in the ward sector that got me if it was something bad. Some could make me angry but I have realized that with years of commitment, looking at the fact that I am a leader makes me relax when things are getting too tough and can think well before I react to the situation. So, I would say that years of work have given me some characteristics or special experience to deal with situations"*. (CNM08)

*"As you keep working with the same patient for a long period, your experience becomes an inner energy that carries you along in the work"* (CNM17)

Other participants described how their years of experience have made them knowledgeable and helped them react to situations differently;

*"...the more I work, the more I became knowledgeable and so it shouldn't be a problem dealing with the kind of patients I care-for now"* (CNM12)

*"Yes, yes it has. My years of experience have thought me a lot. It has changed the way I view things. Now things don't get into me like first due to experience"* (CNM13)

*"Even though I am not trained as a cancer nurse, now I have great knowledge in this area because of work experience"* (CNM 18).

## Inherent desire to help/care for the patients

Again, other participants expressed that their inherent desire to want to help care for their patients was a major factor that promoted resilience. They claimed that most of their colleagues too had this inherent desire to render care for the patients despite the inadequate material and human resources for care. The participants claimed this influenced team building and

that being part of the team desirous of helping patients was useful in helping them to bounce back in stressful situations. Below are some quotes from the participants:

*"What help us is that, when we come to work, we entertain ourselves because we know that no matter what we have to do the work, so the staff-to-staff relationship helps".* (CNM07)

*"With this work, you need to learn to work with others so that you can meet the needs of the patients"* (CNM20)

*"sometimes what keeps you going is the team members who are all interested in helping the patient despite the inadequate personnel and materials for work"* (CNM 01)

*"Hmmmm, you can come to work and the things that you need for work is so limited but you try your best to give the best of care to your patients and we try to encourage each other as we work"* (CNM 17)

*"Is the teamwork here that also keeps us going. So at least we have each other's support"* (CNM13)

*"There is nothing you cannot do if you recognize that you are a team and that everyone must help to make the patient better, this keeps me going* (CNM19)

*"When you are down, your colleagues encourage you and this helps you cope with the work because we try to happy-ourselves"* (CNM 08)

## Maintaining professional outlook

Maintaining a professional outlook was reported as an individual characteristic that the participants employed as a way of building resilience.

*"You try to render your service to the professional standard and take your emotions and feelings out of the work, this helps a lot"* (CNM 15)

*"The work here is so stressful and demanding but we are professionals, so we try to ensure we utilize our professional skills in doing the job, we also encourage each other and help develop our individual professional skills, these helps to promote emotional strength"* (CNM 14)

*"Personally, what has helped me is, I have tuned my mind that when I come to work, I will do my best for my patients, but I will not kill myself. When I leave the ward, I leave everything behind. Forget about everything . . ."* (CNM03)

*". . .I have prepared my mind that when I come to work and I must do what I must do, at the end of the day its 8 hours interval, I will hand over and go. I do what am supposed to do because I get paid"* (CNM02)

*"As for me, I try to stay professional and keep the problems of the workplace at the workplace, I don't want to carry the issues from the workplace to the house and this has helped me a lot"* (CNM 08)

*"Professionalism is our guide, we try to keep to the standard training required and we try our best to deliver quality care to the best of our skill"* (CNM 18)

### Emotional numbness

Some participants narrated that they have lost compassion towards patients' deaths due to frequent exposure to death. The participants expressed emotional numbness as a defense mechanism to keep them from experiencing further psychological or physical suffering. Below are some quotes from the participants' narratives.

> "*At first when a patient dies, I will feel so sad and even cry at times. And sometimes find it very difficult to sleep after performing the last offices but now hmmm this ward, every blessed day people die, so now that compassion is no more. Death is normal to me now*". (CNM10)

> "*We have seen so many dying in this unit that we don't care about deaths anymore*" (CNM 02).

> "*Working for seven years in a ward that most of those who come with advanced cervical cancer die is so serious and makes us emotionally unstable and it seems we have even lost our compassion and feelings*" (CNM 09).

> "*Some of us are used to death right now. When I hear someone is dead, I don't panic anymore, is normal to me . . . So, at first, when we lose a patient, I will be sad the whole day and even end up crying but not anymore. That sympathy is no more due to the frequent deaths*". (CNM13)

> "*But working here hmmm. . . because of the death you feel normal when you hear someone is dead. Yeah! you feel normal . . .*". (CNM01)

> "*Some of us are now used to death, the pain of death doesn't get into me anymore*". (CNM14)

### Workplace characteristics affecting resilience among the nurses and midwives

Regarding the workplace factors influencing resilience, we identified the main theme of demands of caregiving with the following sub-themes emerging; other cases being managed, severity of cases managed, nature of care rendered, activities of care given, reshuffling, schedules and gender mirroring as an exacerbator of psychological suffering.

### Other cases managed

It was discovered in this study that participants do not only manage cervical cancers only but other gynecological cases as well and they claimed that made them not to concentrate on one condition, that is cervical cancer, and they were of the view that this affected the care given to the advanced cervical cancer patients. Some of the participants quotes are below:

> "*The problem is that we have other disease conditions that we manage here and not just what you are studying that is cervical cancer so we don't over concentrate on cervical cancer and this affects the attention given to the patient. . .*". (CNM01)

> "*We manage different conditions on this unit in addition to cervical cancer and this places a huge emotional and work-related burden on us, it's not an easy job especially with the advanced stage cancers who come with a lot of complications*" (CNM 12)

> "*Some different diseases are admitted to this unit, and we have to manage them as well so you cannot concentrate on cervical cancer patients alone and leave the other patients, but these*

*cervical cancer patients also need a lot of attention because of the advanced state that most of them come to the hospital with*" (CNM14)

"*Yes! Have been working here for some time now. This is a gynecological unit and so we deal with gynecological conditions. Ok, hmmm, we have AUBL, Degenerative uterine fibroids), we have endometrial CA, cervical CA, ovarian malignancy, ectopic gestation*" (CNM05)

However, participants revealed that the most common condition they manage is cervical cancer in their advanced forms.

"*We have cervical cancer patients here, endometrial cancers, ovarian cancers but the cervical cancer cases are a lot and they mostly come when the condition has advanced, I mean it has already spread to other organs so the best done for them is palliative care. Which means we manage their symptoms and nurse them to a peaceful death*" (CNM16).

"*The cervical cancer patients come in bad state as compared to the usual disease conditions*" (CNM 11)

"*Most of our cases in this ward is cervical cancer and most of them have gone round for treatment from several places before eventually coming to this hospital in the most advanced state expecting magical transformation, this is what drains us the most*" (CNM 06)

## Severity of cases

The participants revealed that the severity of the patients' condition had a major effect on their psychological state and resilience. They claimed most of the patients reported late to the facility with advanced stage of cervical cancers and this made care rendering very difficult as the complicated cases required total nursing care which affected their level of resilience. This is supported by the quotes below.

"*Most of our cases in this ward are at the end stage or advanced stage so they depend on us the nurses and midwives for everything and cannot assist themselves at all*"(CNM 6)

"*We have the worst form of the conditions in this hospital because all referrals end up here and the same applies to cervical cancer cases. They come in advanced stages with severe symptoms, this pushes us to the extreme and deplete all our emotional reserves*" (CNM 14)

"*hmmmmm, sometimes the patients symptoms are so severe that, you lose hope and this makes you emotionally sick*". (CNM 3)

"*The women who are here are in the advanced stage of the sickness and this makes caring for them demanding on us the nurses and midwives*" (CNM 12)

## Nature of care

The participants revealed that the nature of care given to their patients is palliative care hence nursing them to a peaceful death which places huge demands on the participants and affects their resilience. Below are some of the participants quotes:

"*As for the work, hmmm, most of the patients we deal with are cancer patients, terminally ill patients thus those patients, most of them cannot do anything for themselves, especially their basic self-care needs. Some of them can do certain things compared to others, whilst for others,*

*we have to do everything for them. And once they are here, they are like a family, we do every-thing with them".* (CNM07)

*"With the nature of our work, we have come to terms that; it is a condition that is difficult to battle. And so, we nurse them to die in peace" (CNM 05).*

*"Most of the cases that come here are cases we are nursing to give a peaceful death to, basically palliative care. We are not also trained in that area officially, but we try our best to help"* (CNM10)

*"The patients here require total nursing care, we have to do everything for them and this drains us emotionally"* (CNM 01)

*"When you come to work in this unit, you get home tired, worn-out, emotionally drained and rejected due to the pressure and demands of the work"* (CNM 16)

Again, the participants stated that the kind of activities of care given to their patients forms part of workplace characteristics influencing resilience.

*"We give them all the health care, every care from self-care needs to medication, bathing and all that to the cancer patient. So that is what we do here".* (CNM04)

*"We bathe them, groom them, change their diapers, change their pads because some bleed a lot, you change the next minute is soiled, some vomit too".* (CNM06)

## Reshuffling

It was discovered that nurses and midwives working in this setting require a change of ward after some time to build their resilience. The participants claimed that working on the same unit for a long period led to the reduction of resilience.

*"So, I would prefer that they should be changing us yearly, do you understand? The way the condition of the ward is, they shouldn't keep a nurse here for more than a year or two. For 2 years that's okay, that's if it goes to the extreme. So that, the psychological and that kind of traumatizing moment we have been having all the time at least will reduce so I would prefer that every year or yearly they should do some changes for the ward"* (CNM06)

*"Sometimes we need a change of ward to help us build resilience".* (CNM14)

*"Working for too long on this unit can demoralize you and reduce your psychological strength"* (CNM20)

## Gender mirroring as an exacerbator of psychological suffering

The participants expressed serious worry about being females and managing female-specific conditions. They contended that as women working in female-specific condition in its advanced form took a mental toll on them and contributed to depleting their resilience as they felt they could also suffer the condition in the not-too-distant future.

Below are some quotes from some of the participants.

*"We are also women, so it gives us some sort of anxiety especially when the normal processes of the body are not working well. . .. There is a deviation. Maybe when your menses become*

*irregular you have the anxiety that what is going on and because you are in a gynecology ward, you know that it could be the beginning of something or a big problem*" (CNM04)

"*Sometimes, you feel so worried that your colleague woman is suffering with this condition, and it makes you lose interest in life*". (CNM07)

"*I feel that this condition is so draining on us women and it seems most of us will get it and this makes me worried even after work*" (CNM 19)

"*All of us on this unit are women and you see your colleague women suffering and crying and this goes deep into you so much*" (CNM17]

## Discussion

This study found that the resilience of the participants was influenced by personal factors such as experience as a safety toolkit, inherent desire to help/care for the patient, emotional numbness and maintaining professional outlook. This finding is in line with the literature [4, 6, 10, 17]. As indicated in the study by Matheson and his colleagues [17] that the personal characteristics of work experience and intrinsic passion to render care had significant influences on resilience of hospital staff as they contributed to maintaining a good psychological health.

The participants stated that experience was a safety toolkit in developing and improving their resilience at the workplace. They reported that though they were not specialized in oncology and palliative case, they were of the firm conviction that they were able to deliver high-quality care because of their extensive work experience caring for patients with advanced cervical cancer. Longer years of experience has been identified as a useful tool in building knowledge in clinical settings and this tends to promote resilience [17]. The finding of this study is consistent with other prior studies [18–22] that found that on the job experience increases task competence and quality of care. However, some authors [17, 24] reiterate that experience should not lessen the need for specialized training. It is emphasized that experience enhances skills of clinicians with the requisite specialized training in any domain of health care particularly among nursing staff [21, 23].

Another finding of this study was the finding of inherent desire to help/care for patient with advanced cervical cancer which increased the resilience of the participants. They acknowledged that even though they had little or no support from their superiors, their ability to work with colleagues and other professional team members enhanced their psychological state and influenced their level of resilience. The participants claimed that having the inherent desire to render care was an intrinsic motivation to provide care and this could enhance their resilience [18, 19]. The innate desire to render care among nursing and midwifery staff promotes collaborative care as well as teamwork amid limited resources. Teams provide a supportive nest for health care professionals and help in skill and knowledge sharing among members to promote professional performance and resilience [19, 21]. Teamwork also helps in reducing workplace stress by promoting communication and supportive care [20]. Hence, teamwork and knowledge-sharing should be encouraged as a key factor in promoting professional resilience among nurses and midwives caring for patients with advanced cervical cancer in resource-limited settings.

The study further revealed that the nurses and midwives indicated that they intentionally set boundaries in their professional lives and work to maintain professional outlook. This is done by ensuring that when they are at post, they ensure all their responsibilities have been completed and once off duty, that ends their professional responsibility. While respecting it as a duty that they were being paid to provide, they, however, were keen on avoiding burnout

due to the overwhelming nature of the work and hence employed this strategy as a way of improving their psychological health through resilience. Like this, the literature [5, 7, 9] encourages professional emotional boundaries to be set to promote resilience. The literature suggests emotional boundary setting such as avoiding excessive client connection, isolating work from home or family life, and finding closure after a client passes away as keyways to promote resilience [5, 7, 9]. Having work boundaries or professional boundaries helps to regulate work-family balance and a useful tool in improving mental health [7, 9].

The study also revealed that some of the nurses and midwives had become emotionally numb due to over exposure to patients' pain and frequent death occurrences in their line of work. The participants reported that they were used to their patients' conditions due to years of working with such patients. This led to the loss of compassion and empathy. This finding is consistent with the literature [8, 18, 20]. Some authors [1, 18, 21] argue that when it comes to oncology nursing, workplace emotional numbness develops overtime and there is the need for developing the psychological reserves of staff to improve their mental health [22]. While this method of defense strategy may be useful, if care is not taken, health workers may become compromised in their duty since their sense of compassion and empathy would be lost and this could negatively influence care. This finding concurs with another study conducted by Portoghese et al. [23], who also realized that healthcare professionals who were constantly caring for patients in hospice settings and watching them suffer developed emotional fatigue and moderate burnout.

The mixing of the patients with advanced cervical cancer with other patients with different conditions apart from cervical was seen as detrimental by the participants. There is the need to ensure that cancer patients are managed in designated settings for only such cases and the management should be by specialized clinical team members [24–26]. Unfortunately, almost all the cervical cancer cases reported from this study in the advanced stages of the condition and required specialized care, the participants claimed they were not specialist and only tried rendering specialized care based on their working experience. There is therefore the need for staff with an interest in oncology and or palliative care nursing to be enrolled to specialize in this area of nursing to ensure quality care for cervical cancer patients in Ghana. As can easily be inferred, the participants were providing more of general nursing upgraded specialized care as they had not been specifically trained to provide specialized care. Meeting the specialized care responsibilities towards the patients with advanced cervical cancer can be exhausting and hence the need to have specialists' nurses and midwives trained for the purpose. As stated by some authors [25–28], the psychological competency of nurses and midwives must be enhanced through training in oncology and palliative nursing to effectively meet the comprehensive needs of cancer patients.

The study showed most of the cases managed by the nurses and midwives were terminally ill patients as they reported in the invasive/advanced stage of the condition. The severity of cervical cancer patients' condition made it impossible for a lot of the patients to care for themselves without any kind of assistance. This finding is like those found by [29, 30]. The nature of the care rendered and the activities of care given to the patients places huge job demands on the nurses and midwives and predisposed them to low resilience as reported earlier [4, 22, 29].

Similarly, the study identified that being engaged on the same unit for long time without reshuffling led to low resilience and contributed to the participants likelihood of experiencing low resilience and burnout. Some studies [4, 7, 23] have reported that working in terminally ill setting without change in environment resulted in depletion of emotional reserves and predisposed employees of those settings to increased burnout due to decrease in the level of resilience over time. We recommend tactfulness by nurse managers when handling the placement and reshuffling of nurses and midwives in oncology settings to ensure the emotional reserves of

nurses and midwives are not depleted while still building the requisite work experience in each nurse or midwife to render quality nursing care.

Further, the study found gender mirroring as an exacerbator of psychological suffering among the participants. Some participants expressed that being females and managing female-specific conditions made them anxious and this affected their level of resilience and psychological state. They contended that as women working in female-specific conditions in its advanced forms, they were emotionally unstable and contributed to depleting their resilience as they felt they could also suffer the condition in the not-too-distant future. The participants saw themselves in the "patients-shoes" and this took an emotional toll on the participants predisposing them to low resilience. Other studies [8, 22–26, 30] have stated that caring for terminally ill leads to depletion of emotional resources especially when the care giver identifies himself/herself with the patient being catered for or the patients' condition especially in advanced stages of cancer where the patient is in the terminal stage of life. This calls for structured methods of promoting empathy among the participants so that they reduce the tendency to be sympathetic instead of empathetic. Empathy training and practice could promote resilience just as psychological counselling [28, 30].

## Limitations of the study

Only twenty nurses and midwives at the Korle Bu Teaching hospital were studied and thus difficult to generalize the findings for all nurses and midwives caring for advanced cervical cancer even though it provides deep insights for policy decision making. Comparative studies with similar populations in other settings are encouraged to help broaden the literature in the study area.

The study participants were general nurses and midwives working in the national referral hospital with none of the participants having specialized training in oncology or palliative care. Specialist nurses and midwives could be more resilient than the general nurses and midwives studied, future studies could look at a comparison between general nurses and midwives with specialized nurses and midwives.

The findings of the study could have been affected by social desirability as the participants could have given responses to suit the social environment as with most interview-based studies.

## Conclusion

Resilience among nurses and midwives caring for terminally ill cervical cancer patients is influenced by longer years of service, intrinsic motivation to work as a nurse, and the defense strategy of emotional numbness and professionalism at the individual level. Also, the huge demand of caregiving serves as a major workplace factor affecting the resilience of nurses and midwives.

We recommend strategies such as regular ward conferences and in-service trainings aimed at enhancing job-experience, inherent desire to render care and professionalism be adopted in resource-constrained settings to improve nurses' resilience. In addition, political actors and management of hospitals must prioritize allocation of resources for advanced cervical cancer care with particular focus on providing more specialized nurses and midwives.

## Supporting information

**S1 Appendix. Interview guide.**
(DOCX)

## Acknowledgments

Sincere appreciation to the study participants for taking part in this study for the advancement of oncology nursing science and palliative care.

## Author Contributions

**Conceptualization:** Jennifer Oware, Merri Iddrisu, Gladys Dzansi.

**Formal analysis:** Jennifer Oware, Kennedy Dodam Konlan.

**Investigation:** Merri Iddrisu, Kennedy Dodam Konlan, Gladys Dzansi.

**Methodology:** Jennifer Oware, Merri Iddrisu, Gladys Dzansi.

**Validation:** Kennedy Dodam Konlan.

**Writing – original draft:** Kennedy Dodam Konlan, Gladys Dzansi.

**Writing – review & editing:** Jennifer Oware, Merri Iddrisu.

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
