## [Decision Letter · Decision Letter 0]

30 Sep 2024

PONE-D-24-31816Personal and workplace factors influencing the resilience of nurses caring for women with cervical cancer in a resource-constrained setting in GhanaPLOS ONE

Dear Dr. Konlan,

Thank you for submitting your manuscript to PLOS ONE. After careful consideration, we feel that it has merit but does not fully meet PLOS ONE’s publication criteria as it currently stands. Therefore, we invite you to submit a revised version of the manuscript that addresses the points raised during the review process.

We look forward to receiving your revised manuscript.

Kind regards,

Collins Atta Poku

Academic Editor

PLOS ONE

Journal Requirements:

1. When submitting your revision, we need you to address these additional requirements. Please ensure that your manuscript meets PLOS ONE's style requirements, including those for file naming. The PLOS ONE style templates can be found at https://journals.plos.org/plosone/s/file?id=wjVg/PLOSOne_formatting_sample_main_body.pdf and https://journals.plos.org/plosone/s/file?id=ba62/PLOSOne_formatting_sample_title_authors_affiliations.pdf 2. Please provide a complete Data Availability Statement in the submission form, ensuring you include all necessary access information or a reason for why you are unable to make your data freely accessible. If your research concerns only data provided within your submission, please write "All data are in the manuscript and/or supporting information files" as your Data Availability Statement. 3. PLOS requires an ORCID iD for the corresponding author in Editorial Manager on papers submitted after December 6th, 2016. Please ensure that you have an ORCID iD and that it is validated in Editorial Manager. To do this, go to ‘Update my Information’ (in the upper left-hand corner of the main menu), and click on the Fetch/Validate link next to the ORCID field. This will take you to the ORCID site and allow you to create a new iD or authenticate a pre-existing iD in Editorial Manager. 4. In the online submission form, you indicated that "Data is available upon request from the corresponding author". All PLOS journals now require all data underlying the findings described in their manuscript to be freely available to other researchers, either 1. In a public repository, 2. Within the manuscript itself, or 3. Uploaded as supplementary information.This policy applies to all data except where public deposition would breach compliance with the protocol approved by your research ethics board. If your data cannot be made publicly available for ethical or legal reasons (e.g., public availability would compromise patient privacy), please explain your reasons on resubmission and your exemption request will be escalated for approval. 5. Your ethics statement should only appear in the Methods section of your manuscript. If your ethics statement is written in any section besides the Methods, please move it to the Methods section and delete it from any other section. Please ensure that your ethics statement is included in your manuscript, as the ethics statement entered into the online submission form will not be published alongside your manuscript. 6. Please include captions for your Supporting Information files at the end of your manuscript, and update any in-text citations to match accordingly. Please see our Supporting Information guidelines for more information: http://journals.plos.org/plosone/s/supporting-information.

Reviewers' comments:

Reviewer's Responses to Questions

**Comments to the Author**

1. Is the manuscript technically sound, and do the data support the conclusions?

Reviewer #1: Yes

Reviewer #2: Yes

2. Has the statistical analysis been performed appropriately and rigorously? 

Reviewer #1: N/A

Reviewer #2: Yes

3. Have the authors made all data underlying the findings in their manuscript fully available?

Reviewer #1: No

Reviewer #2: Yes

4. Is the manuscript presented in an intelligible fashion and written in standard English?

Reviewer #1: Yes

Reviewer #2: Yes

5. Review Comments to the Author

Reviewer #1: The authors have presented an important research article with the aim of exploring personal and workplace factors influencing the resilience of nurses caring for patients with advanced cervical cancer in a resource constrained setting.

The methodology was well presented however, the authors needs to address the following observations:

1. In the data collection section, it was stated that 64 nurses and midwives were contacted but only 48 expressed interest in participating and only 20 participants were actually interviewed. Were this 20 selected from the 48 after meeting the inclusion criteria?

2. And is 64 the total number of nurses and midwives that work in the unit where the participants were recruited?

3. In the subsection titled 'Rigour', in the first line the authors mention that "the interview guide was pretested among two patients...". Were patients also involved or is this a typographical error?

4. During the interview, were the participants asked the same questions in the same format or did these vary according to participants?

Reviewer #2: Questions to authors

1. What criteria were used to determine data redundancy and and data saturation after the sixteenth interview? And why was it decided to add exactly four respondents?

2. Did authors have criteria for estimation of poor reporting culture of patients?

3. Did authors used participant information sheet in local language ?

6. PLOS authors have the option to publish the peer review history of their article (what does this mean?). If published, this will include your full peer review and any attached files.

Reviewer #1: No

Reviewer #2: **Yes: **Raikhan Bolatbekova

---

## [Author Response · Author response to Decision Letter 0]

7 Oct 2024

University of Ghana

College of Health Sciences

School of Nursing and Midwifery

Department of Adult Health

1st October, 2024

The Editor

PLOS ONE

Dear Sir/Madam,

RESPONSE TO REVIEWERS

We are grateful for the review of our manuscript titled “Personal and workplace factors influencing the resilience of nurses caring for women with cervical cancer in resource constained setting in Ghana” to make it better for possible publication.

Comments from the editor

Journal Requirements:

4. In the online submission form, you indicated that "Data is available upon request from the corresponding author".

Response to editor

1. We have complied with the requirements for file naming as suggested by the editor.

2. We have complied with the suggestions on data availaibility by providing the recommended data availability statement in the manuscript. This is on page 23 of the revised manuscript. 

3. The ORCID iD for the corresponding author is already part of the information for the Editorial Manager.

4. We have revised the data availability statements in the submission system as suggested by the editor.

5. We have provided the ethics section in the methods section of the manuscript as suggested by the editor . This is on page 10 of the revised manuscript. 

6. We have provided the appropriate captions for the supporting information as suggested by the reviewer and same has been provided at the end of the manuscript. 

7. We have reviewed the reference list and ensure they comply with the Journal’s specifications. 

Reviewers' comments:

Reviewer's Responses to Questions

Comments to the Author

1. Is the manuscript technically sound, and do the data support the conclusions?

Reviewer #1: Yes

Reviewer #2: Yes

2. Has the statistical analysis been performed appropriately and rigorously? 

Reviewer #1: N/A

Reviewer #2: Yes

3. Have the authors made all data underlying the findings in their manuscript fully available?

Reviewer #1: No

Reviewer #2: Yes

4. Is the manuscript presented in an intelligible fashion and written in standard English?

Reviewer #1: Yes

Reviewer #2: Yes

Review Comments to the Authors

Reviewer #1: 

The authors have presented an important research article with the aim of exploring personal and workplace factors influencing the resilience of nurses caring for patients with advanced cervical cancer in a resource constrained setting.

The methodology was well presented however, the authors need to address the following observations:

1. In the data collection section, it was stated that 64 nurses and midwives were contacted but only 48 expressed interests in participating and only 20 participants were interviewed. Were this 20 selected from the 48 after meeting the inclusion criteria?

2. And is 64 the total number of nurses and midwives that work in the unit where the participants were recruited?

3. In the subsection titled 'Rigour', in the first line the authors mention that "the interview guide was pretested among two patients...". Were patients also involved or is this a typographical error?

4. During the interview, were the participants asked the same questions in the same format or did these vary according to participants?

Responses of Authors to Reviewer 1

1. Yes, the twenty participants were selected from the eligible forty-eight participants who were ready to participate in the study. This has been explained on page 6 of the revised manuscript. 

2. The 64 were the nurses and midwives at the Unit who met the inclusion criteria and not the total number of nurses working at the unit. This is stated on page 6 of the revised manuscript on data collection.

3. We have rectified the typographical error in the first sentence of the rigour section. The study tool was pre-tested among nurses. This is stated on page 9 of the revised manuscript.

4. We have indicated that the participants were sasked the same questions in the same format with further probing questions to elicit in-depth information. This stated on page 7 of the revised manuscript.

Reviewer #2: 

Questions to authors

1. What criteria were used to determine data redundancy and data saturation after the sixteenth interview? And why was it decided to add exactly four respondents?

2. Did authors have criteria for estimation of poor reporting culture of patients?

3. Did authors used participant information sheet in local language ?

Responses of Authors to Reviewer 2

1. The criteria for data redundancy and data saturation has been stated on page 6 of the revised manuscript specifically on the continuation of study population and sample size determination. We have provided our reasons for the addition of the four additional interviews after the 16th interview on page 6 of the revised manuscript.

2. Respectfully, the study was not on patients and we did not collect information on poor reporting culture of patients. This can be explored in future studies.

3. We have indicated that we used participant information sheet in English since all the participants were literate. This has been stated on pages 6 and 10 of the revised manuscript.

Thank you 

Yours sincerely,

Dr. Kennedy Dodam Konlan 

(Corresponding author)

---

## [Decision Letter · Decision Letter 1]

18 Nov 2024

Personal and workplace factors influencing the resilience of nurses caring for women with cervical cancer in a resource-constrained setting in Ghana

PONE-D-24-31816R1

Dear Dr. Dodam Konlan,

We’re pleased to inform you that your manuscript has been judged scientifically suitable for publication and will be formally accepted for publication once it meets all outstanding technical requirements.

Kind regards,

Collins Atta Poku

Academic Editor

PLOS ONE

Additional Editor Comments (optional):

Reviewers' comments:

Reviewer's Responses to Questions

**Comments to the Author**

1. If the authors have adequately addressed your comments raised in a previous round of review and you feel that this manuscript is now acceptable for publication, you may indicate that here to bypass the “Comments to the Author” section, enter your conflict of interest statement in the “Confidential to Editor” section, and submit your "Accept" recommendation.

Reviewer #1: All comments have been addressed

Reviewer #2: All comments have been addressed

2. Is the manuscript technically sound, and do the data support the conclusions?

Reviewer #1: Yes

Reviewer #2: Yes

3. Has the statistical analysis been performed appropriately and rigorously? 

Reviewer #1: Yes

Reviewer #2: Yes

4. Have the authors made all data underlying the findings in their manuscript fully available?

Reviewer #1: Yes

Reviewer #2: Yes

5. Is the manuscript presented in an intelligible fashion and written in standard English?

Reviewer #1: Yes

Reviewer #2: Yes

6. Review Comments to the Author

Reviewer #1: (No Response)

Reviewer #2: Authors made all changes based on the recommendations and answered to all questions

Abstract

Clearly study objective is provided

Introduction

the authors fully revealed the relevance of the topic globally and locally in Ghana.

Objective and aim are clear

Methods

IRB approved study

There was clearly and detailed information how data was collected :

- exclusion and inclusion criteria described clearly in chapter

-rmation about addition of participants.

Results

Data performed chronologically

There was no figures/tables in the chapter

Discussion

1. Chapter is started from concise statement summarizing the main findings of the study

“This study found that the resilience of the participants was influenced by personal factors such as experience as a safety toolkit, inherent desire to help/care for the patient, emotional numbness and maintaining professional outlook.”

2. Authors described and compared similar studies with similar objectives and results

3. Authors presented weakness and strengths of study

4. Highlighted statement what is offered by the study for future direction

Authors consider alternative interpretations and proffer practical implications.

References

Most of the references is no more than10 years

Quality of English language

The level of the English is high

7. PLOS authors have the option to publish the peer review history of their article (what does this mean?). If published, this will include your full peer review and any attached files.

Reviewer #1: No

Reviewer #2: **Yes: **Raikhan Bolatbekova

---

## [Editor Report · Acceptance letter]

19 Nov 2024

PONE-D-24-31816R1 

PLOS ONE

Dear Dr. Konlan, 

I'm pleased to inform you that your manuscript has been deemed suitable for publication in PLOS ONE. Congratulations! Your manuscript is now being handed over to our production team.

Kind regards, 

on behalf of

Dr. Collins Atta Poku 

Academic Editor

PLOS ONE